# Multi-Criteria Usability Evaluation of mHealth Applications on Type 2 Diabetes Mellitus Using Two Hybrid MCDM Models: CODAS-FAHP and MOORA-FAHP

**Kamaldeep Gupta [1], Sharmistha Roy [1], Ramesh Chandra Poonia [2], Raghvendra Kumar [3], Soumya Ranjan Nayak [4], Ayman Altameem [5] and Abdul Khader Jilani Saudagar [6,*]**

[1] Faculty of Computing and Information Technology, Usha Martin University, Ranchi 835103, India; kamal.sxcranchi@gmail.com (K.G.); sharmistha@umu.ac.in (S.R.)
[2] Department of Computer Science, CHRIST (Deemed to be University), Bangalore 560029, India; rameshchandra.poonia@christuniversity.in
[3] Department of Computer Science and Engineering, GIET University, Rayagada 765022, India; raghvendra@giet.edu
[4] Amity School of Engineering and Technology, Amity University Uttar Pradesh, Noida 201303, India; srnayak@amity.edu
[5] Department of Computer Science and Engineering, College of Applied Studies and Community Services, King Saud University, Riyadh 11533, Saudi Arabia; aaltameem@ksu.edu.sa
[6] Information Systems Department, Imam Mohammad Ibn Saud Islamic University (IMSIU), Riyadh 11432, Saudi Arabia
[*] Correspondence: aksaudagar@imamu.edu.sa





**Simple Summary:** Considering the world's present pandemic situation, mHealth applications are essential for self-management of health. However, the recent growth of the healthcare industry provides various mHealth applications that result in difficulties for ordinary people in selecting the best application to fulfill their needs to their satisfaction. This research paper focuses on developing two hybrid decision-making methods, CODAS-FAHP and MOORA-FAHP, which may be used in assessing the usability of mHealth applications to monitor type 2 diabetes mellitus (T2DM) patients. The resulting analysis with the two hybrid models shows that the selection of mHealth applications can be done efficiently.

**Abstract:** People use mHealth applications to help manage and keep track of their health conditions more effectively. With the increase of mHealth applications, it has become more difficult to choose the best applications that are user-friendly and provide user satisfaction. The best techniques for any decision-making challenge are multi-criteria decision-making (MCDM) methodologies. However, traditional MCDM methods cannot provide accurate results in complex situations. Currently, researchers are focusing on the use of hybrid MCDM methods to provide accurate decisions for complex problems. Thus, the authors in this paper proposed two hybrid MCDM methods, CODAS-FAHP and MOORA-FAHP, to assess the usability of the five most familiar mHealth applications that focus on type 2 diabetes mellitus (T2DM), based on ten criteria. The fuzzy Analytic Hierarchy Process (FAHP) is applied for efficient weight estimation by removing the vagueness and ambiguity of expert judgment. The CODAS and MOORA MCDM methods are used to rank the mHealth applications, depending on the usability parameter, and to select the best application. The resulting analysis shows that the ranking from both hybrid models is sufficiently consistent. To assess the proposed framework's stability and validity, a sensitivity analysis was performed. It showed that the result is consistent with the proposed hybrid model.

**Keywords:** usability; T2DM mHealth applications; FAHP; CODAS; MOORA; usability score

# 1. Introduction

Several healthcare professionals and organizations who are seeking cost-efficient ways to deliver good quality healthcare services to patients may find mobile devices to be beneficial [1]. Mobile technology uses mHealth (mobile health) applications in providing health benefits to customers [2]. As mobile devices become more prevalent, mHealth applications that support medical treatment are becoming increasingly ubiquitous. Considering the present health scenario, type 2 diabetes mellitus (T2DM) is one of the various chronic diseases suffered by human beings. Establishing a specific treatment plan, regular nutrition counseling, blood glucose (BG) monitoring, and medication administration are all important components of efficient diabetes management [3]. The mHealth applications look promising in the efficient self-management and monitoring of T2DM patients.

The use of mHealth applications, particularly in self-management, is a challenging and time-consuming task [4]. With the rapid growth of several mHealth applications, it becomes difficult for the users to select the best applications. User interface design and user experience plays vital roles in analyzing and determining the best user-friendly mHealth application. Thus, usability evaluation is essential for choosing the best mHealth applications among the available alternatives.

Usability, as defined by International Organization for Standardization 9241-11 [5], is the degree to which a product can be utilized by a given individual in a particular circumstance to accomplish realistic goals related to effectiveness, efficiency, and satisfaction. Usability in diabetes applications refers to the user's (e.g., a patient, a clinician, or a caregiver) experience of an interface for using the application thereby expecting good satisfaction, efficiency for saving time, and adequate efficacy that the application functions smoothly and accurately in a preferred usage context (e.g., tracking and analyzing Blood Glucose and carb intake). Automated blood glucose data transfer, incorporation of a diary for tracking and logging food intake, physical activity, setting reminders, facilitating education, establishing communication, and social networking are all possible features of diabetes applications [6,7].

Identification of the best mHealth application for monitoring T2DM is a challenging issue. MCDM methodologies can address this issue. MCDM is a structured and multidimensional procedure established to address decision-making problems in a variety of fields and helps to find the most appealing alternative by taking into account all relevant criteria. It studies complex decision-making problems in various fields with its powerful and sophisticated tools. This strategy increases decision-making quality by keeping decision-making reasonable and efficient [8]. According to Hwang and Yoon [9], the MCDM procedures have the following common characteristics:

- Multiple criteria: Every problem has a set of criteria, which might be characterized either as objectives or attributes.
- Criteria in conflict: Several criteria contradict or disagree with each other.
- Incommensurable unit: A criterion maybe measured in different units.
- Design/selection: A problem can be solved either by designing the ideal or the best alternative(s) or by choosing the best alternative from the finite alternatives that have already been identified.

The two types of criteria are attributes and objectives. As a result, several authors classify MCDM techniques into two categories, on the basis of the problem situations [10]: Multi-Attribute Decision Making (MADM) and Multi-Objective Decision Making (MODM).

- MADM: For evaluation, a discrete set of decision-making variables (finite alternatives and attributes) are commonly utilized [11]. The best option in a MADM problem is chosen from a group of pre-selected options that are presented in the form of attributes, which are usually in conflict. The major aspect of MADM is that there are generally a few predefined alternatives, each of which is related to a level of attribute achievement. The ultimate selection will be based on the attributes. Inter and intra-attribute comparisons are also used to make the ultimate choice of the alternative.

On the basis of various types and significant features of information obtained from various decision-makers, Hwang and Yoon [9] categorized different typical MADM techniques. One of the commonly used methods involving MADM methodology is combinative distance-based assessment (CODAS).

- MODM: MODM is a type of MCDM with a decision space that is continuous, indicating an infinite set of alternatives and attributes. MODM's purpose is to select the best option from an endless range of available alternatives that are listed below a set of limitations. As a result, an MODM problem entails the creation of alternatives that optimize or best satisfy the decision-makers' objectives. MODM issues are characterized by the fact that decision-makers must achieve many objectives that are incommensurable and that conflict with one another. A MODM model takes into account a matrix of decision-making variables, objective functions, and constraints. The most widely used MODM method is multi-objective optimization on the basis of ratio analysis (MOORA).

Fuzzy set theory is applied to handle ambiguity in the subjective judgment of a decision-maker. The research presented in this study makes two contributions. First, it aids both health care practitioners and consumers in identifying the important factors/criteria that contribute to the usability aspect of mHealth applications. Second, it outlines a technique for selecting the best mHealth applications among the alternatives, using two hybrid models, CODAS-FAHP and MOORA-FAHP. The fuzzy Analytic Hierarchy Process (FAHP) is utilized to determine the weights of the criteria by removing human vagueness, while CODAS and MOORA are used for determining the different ranks associated with the alternatives.

The aims of this proposed research paper are mentioned below:

- Identification of criteria and sub-criteria associated with the usability feature of mHealth applications.
- Computing criteria weight using the FAHP by removing vagueness and ambiguity in expert judgments.
- Demonstrating two hybrid MCDM model for ranking the alternatives on the basis of the usability scores obtained across multiple criteria and sub-criteria.
- Performing a sensitivity analysis to check the consistency of the result obtained by the proposed hybrid models.

The remaining parts of this paper are organized as follows: a literature review in Section 2 focuses on the usability aspects of the MCDM approaches in the selection of the methodology; then, proposed hybrid MCDM approaches for selecting the best mHealth application, based on usability, are presented in Section 3; Section 4 discusses the methodologies adopted in this paper; the proposed models' result analyses and validations are shown in Section 5; finally, Section 6 provides the conclusion.

## 2. Related Works

Keshavarz et al. [12] proposed a novel CODAS technique to deal with MCDM issues. For assessing the desirability of alternatives, this methodology uses the Euclidean distance (ED) and the taxicab distance (TD) formulae as the primary and supplementary measures, respectively. The negative ideal point is used to calculate both ED and TD. In the CODAS approach, the alternative having the greatest distance is preferred. To solve facility location issues, Kahraman et al. [13] employed four fuzzy multi-attribute group decision-making (FMAGDM) strategies that took into consideration both quantitative and qualitative criteria. They examined the techniques with respect to computational complexity and discovered FAHP to be very challenging. Chen et al. [14] proposed a two-step fuzzy decision-making methodology to locate the warehouse in a supply chain. In a traditional supply chain system involving market demand unpredictability, they examined the simultaneous optimization of the problem related to multiple conflicting objectives. Shuo et al. [15] developed a new FMADM methodology for facility placement difficulties that includes a simple additive

weighting mechanism employing the fuzzy technique. The work was not able to tackle challenges involving many facility sites.

The FAHP model was utilized by SelimZaim et al. [16] to handle the issues of advanced MCDM in the process of selecting the supplier. Ali Nazeri et al. [17] developed a comprehensive technique for searching and evaluating the supplier in the management of the supply chain. Suppliers were evaluated in the first round based on qualitative criteria that included service level, financial health, and loyalty. The weights were determined using the FAHP.

Fuzzy logic and triangular fuzzy numbers (TFNs), on the other hand, are commonly used to deal with the complexities of human judgment. To develop a multi-objective linear programming (MOLP) model, the assessment and evaluation of suppliers are carried out by utilizing quantitative criteria that include cost price, defect rate, and delay, while taking into account afirm's requirements and the suppliers' limitations.

Wang et al. [18] used a hybrid FAHP along with a green DEA model to construct an MCDM framework to deal with durable supplier selection for the manufacturing of edible oil. Depending on the suggestions of the company's procurement professionals, the study used a hybrid MCDM method that incorporated an FAHP, including a GDEA method for determining criteria weights in the procedure for selecting a supplier. The authors of the paper [19] offered the MCDM approach for evaluating and selecting providers of N-hexane solvent (C6H14) in the production of vegetable oil. Experts specified all aspects that impacted the selection of the supplier of the hexane solvent and the inspection process. The criteria weights were then estimated using the FAHP's multi-criteria comparison analysis. Eventually, the technique for order of preference by similarity to ideal solution (TOPSIS) was employed in choosing the optimum provider of the hexane solvent. Ghosh and Roy [20] described how to find the optimum maintenance mix across various components of aplant's manufacturing process by applying a fuzzy decision-making methodology.

Panchal and Kumar [21] investigated how the FAHP and the fuzzy TOPSIS technique may be utilized to find the best strategy related to the maintenance of a thermal power plant's power production unit. Pourjavad et al. [22] used a TOPSIS model based on an analytic network process (ANP) to find the optimal mining maintenance approach. To investigate the viable maintenance option, Fouladgar et al. [23] suggested a technique called fuzzy multiple-criteria decision-making (FMCDM) that depended on a complex proportional assessment (COPRAS) and an analytic hierarchy process (AHP). The assessment criteria weights were generated using the FAHP, and the alternatives' rankings were determined using the COPRAS technique. While evaluating the endpoint of the risk factor stages, Shukri et al. [24] employed FAHP to identify the relative significance of the associated criteria used for the option, which has gained popularity in decision-making settings. Expert subjective judgment was used to generate the priority weight vector with each risk item, using a paired comparison.

For analyzing and choosing a suitable vendor, Shahanaghi and Yazdian [25] developed fuzzy group decision-making utilizing the TOPSIS methodology. Ilangkumaran et al. [26] provided a hybrid MCDM method for solving the issue of selecting a supplier. The FAHP was employed to identify the criteria weights and analyze the design of the problem of supplier selection. Preference ranking organization method for enrichment evaluation (PROMETHEE) was adopted to find the overall rating associated with the providers. For green supply chain supplier selection, Kannan et al. [27] employed fuzzy MCDM, using a model that focused on multi-objective programming. This model aimed to increase the entire value of purchases, thereby lowering the overall cost.

In a fuzzy environment, Wang et al. [28] adapted TOPSIS for fuzzy multiple-criteria group decision-making (FMCGDM). The authors contributed two operators, Up and Lo, to the generalization of TOPSIS, which partially fulfill the ordering relation related to the fuzzy numbers. To choose the optimum effective manufacturing plant maintenance methods, Al-Najjar and Alsyouf [29] used FMCDM and other approaches related to fuzzy reasoning inference. Willem Karel M. Brauers [30] explored the MOORA methodology

and its prospective applications in the area of privatization in welfare-related economies, emphasizing the manufacturing unit and the global competitive market. The MOORA method for ECM process parametric optimization was studied by Vedansh Chaturvedi [31]. To produce the best parametric output, the optimization utilizes some provided input parameters, such as voltages, feed rate, and so on. This method is effective in resolving multi-objective optimization problems and providing process quality enhancement.

Aydomuş et al. [32] created a picture fuzzy sets-based modification of the CODAS approach. A novel approach was presented in this regard to determine how picture fuzzy numbers could be incorporated into the CODAS method. Depending on the Euclidean and taxicab distances, as well as a negative ideal solution, the suggested technique combined multi-criteria decision analysis and picture fuzzy uncertainties. The technique, picture fuzzy CODAS, was used to solve the problem of ERP system selection.

Panchal et al. [33] chose the most sustainable oil for a cleaner and more environmentally friendly manufacturing method in Agra's small-scale casting businesses. The FAHP, the fuzzy technique for order of preference by similarity to ideal solution (FTOPSIS), and fuzzy evaluation based on distance from average solution (FEDAS) approaches were all part of a revolutionary integrated three-phase decision-making paradigm. FAHP was used to calculate the weights of the various criterion and sub-criteria. These weights were then utilized in the FTOPSIS and FEDAS techniques to rank the oils under consideration.

Mohamad Arif et al. [34] suggested a risk-based FAHP technique to review the Android mobile application as part of an MCDM mobile malware detection technique. The objective of this research was static analysis, which employed permission-based aspects to evaluate the mobile malware detection methodology. Risk analysis was used to make the mobile user more aware of the risks involved in approving every authorization demand.

Under a Pythagorean fuzzy environment, Almeraz-Durán et al. [35] created a hybridization of the failure mode and effect analysis (FMEA) and the CODAS methods. The outcomes were then compared to a MOORA methodology-based variant of the suggested model.

At Sekolah Tinggi Akuntansidan Manajemen Indonesia (STAMI), Siregar et al. [36] resolved the difficulty of choosing the top students as recipients of an academic achievement improvement scholarship. STAMI's scholarship data processing was carried out manually; this problem was overcome with the help of a decision support system and the MOORA method. The MOORA approach was used to choose the most deserving students for scholarships.

On the basis of the above literature review, it is obvious that no researcher has yet employed the suggested integrated fuzzy MCDM framework to handle a decision-making issue in any domain. Due to this research gap, integrated frameworks based on the proposed FAHP and two MCDM methodologies for rank determination, CODAS and MOORA, are utilized for selecting the optimal T2DM mHealth application among the five alternatives taken in this research work.

## 3. Proposed Hybrid Models

The authors in this paper have presented hybrid MCDM models for ranking the various mHealth applications and for selecting the best applications. To begin with, a hierarchical structure was built utilizing 10 criteria and 29 sub-criteria that were determined after a review of literature and considering the opinions provided by experts. The FAHP's technique was utilized in this work, since it only requires the construction of a few pair-wise comparison matrices, rendering the method easier and more systematic in its application [37]. The current study utilized hybrid methodologies that combined the FAHP and two MCDM approaches, CODAS and MOORA, for the evaluation of the best T2DM mHealth application.

The methodologies were implemented as follows. The FAHP assessed the relative criteria and sub-criteria weights. These weights were then utilized in the MADM (CODAS) and MODM (MOORA) approaches to rate and rank the multiple mHealth alternatives

within a fuzzy environment, by applying the fuzzy set theory for controlling the unpre-dictability of the evaluation processes. As a result, we concluded that using the FAHP, CODAS, and MOORA, the problem of picking the best T2DM mHealth application could be solved. The two segments of the suggested integrated methodologies for finding the best T2DM mHealth applications are shown in Figure 1.

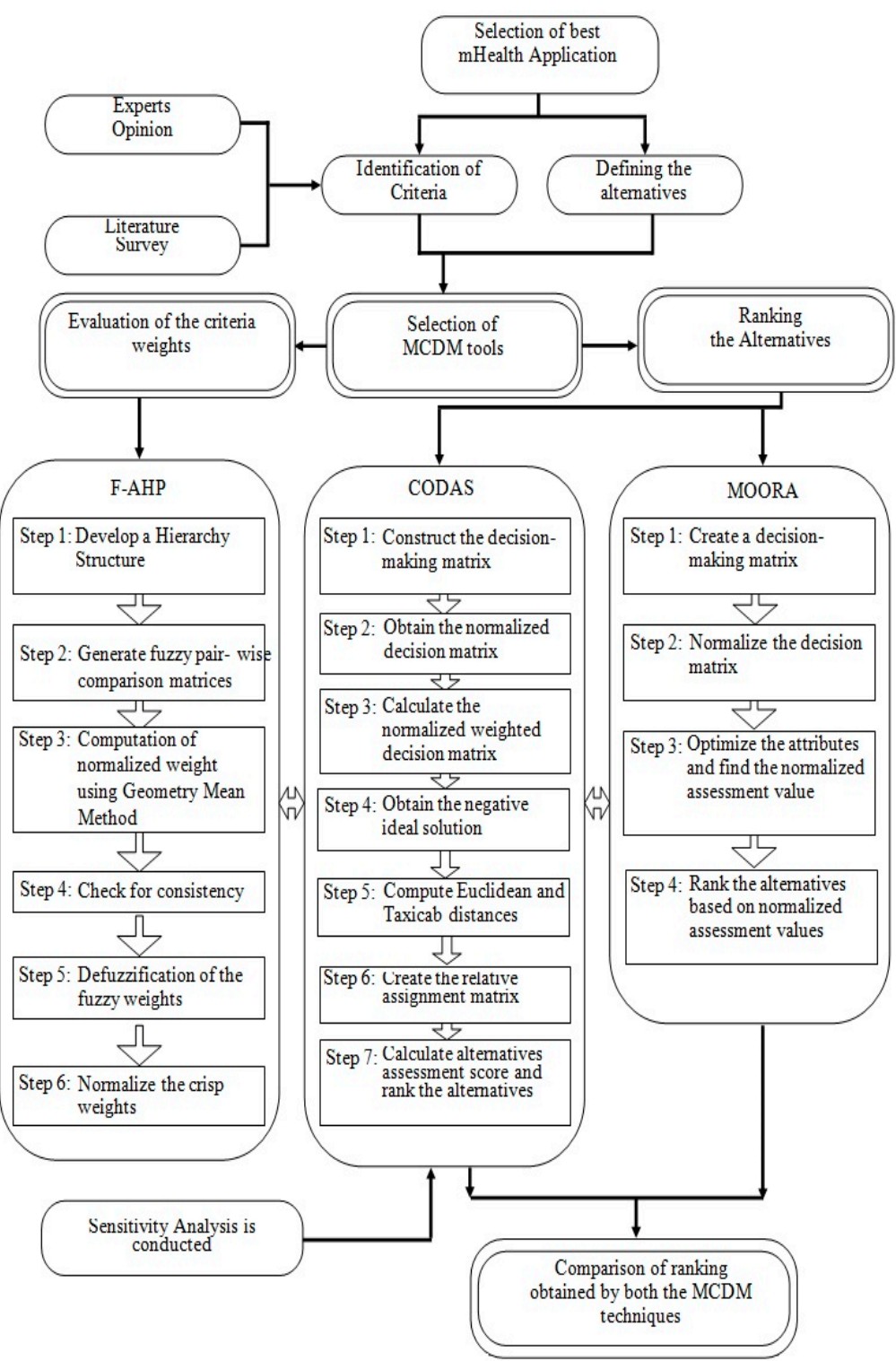

**Figure 1.** Proposed Hybrid Models.

Diverse criteria and their associated sub-criteria were identified during the first segment of the FAHP, using various sources including a literature review and consideration of expert opinion. Expert opinion and pairwise comparison matrices were used to create a decision hierarchy structure using assessed criteria, associated sub-criteria, and choices. The weights related to the criteria and their associated sub-criteria were tabulated by employing the geometric mean (GM) methodology. In the second section, the ranking of the alternatives was accomplished by using the two methodologies, both of which used global weights. Global weights were the product of the criteria weight and associated sub-criteria weight.

The two methodologies are:

- CODAS: The negative ideal solutions were generated in this MADM technique. Using the fuzzy negative solution values, the Euclidean distance (ED) and the taxicab distance (TD) were determined, and the assessment scores ($AS_i$) for each alternative were tabulated. The choices were ranked in descending order. The ranking outcomes' consistency and validity were assessed using sensitivity analysis.

- MOORA: The MODM technique was utilized to create the weighted normalized decision matrix and the normalized assessment values. The alternatives were ranked using the normalized assessment values.

The following pseudo code Algorithm 1 describes the proposed hybrid models for evaluating the usability of T2DM mHealth applications and ranking the best application. The proposed algorithm uses Fuzzy AHP for determination of criteria weights as mentioned in Algorithm 2. Algorithm 3 mentions the CODAS method, used for ranking the alternatives. MOORA method mentioned in Algorithm 4 below is also used for ranking of the alternatives. A comparison study has been conducted between these two hybrid MCDM models for checking the consistency.

---

**Algorithm 1:** Hybrid MCDM Model

---

```
1    {
2    // The following pseudocode is to select the best T2DM mHealth application
3    // among 'n' number of applications based on usability criteria.
4    Input: Alternatives: 'n' number of different mHealth applications. (n∈ℝ and
5    n > 1)
6    Criteria: 'm' number of homogeneous features related to T2DM mHealth
7    applications based on which the alternatives will be evaluated. (m∈ℝ and
8    m > 1)
9    Sub-criteria: 's' number of sub-features for each criteria identified for
10   evaluating usability of T2DM mHealth applications (s∈ℝ and s ≥ 1)
11   Output: Ranking of the alternatives and selection of best T2DM mHealth
12   application based on rank
13   n[1:5, m[1:10], s:=minimum value of s is 1 and maximum value of s is 5
14   for i:=1 to m do
15   cr_weight[i]:=F_AHP(m[i], s)
16   for i:=1 to n do
17   AS[i]:=CODAS (cr_weight, n[i]);
17   // Ranking of the Alternatives is done based on Assessment Score. Higher the
18   // AS[i] value the higher is the rank
19   for i:=1 to n do
20   NAV[i]:=MOORA (cr_weight, n[i]);
21   // Ranking of the Alternatives is done based on NAV. Higher the NAV[i]
22        // value, the higher is the rank
23   }
```

---

**Algorithm 2:** F_AHP(m, s)// Fuzzy AHP

---

```
1   {
2   // The following pseudocode will evaluate the criteria and sub-criteria for
3      // Consistencyevaluation and determination of weights
4   Input: Criteria ('m' number of features) and Sub-criteria ('s' number of sub-
5   features for each criteria)
6   Output: Calculation of Criteria weights, sub-criteria weights and global
7   weights.
8   i. Generating the fuzzy pairwise comparison matrices based on fuzzy conversion
9   scale
10         ii. Computation of nrmalized weight values using Geometric Mean (GM)
11  method
12             iii. Check for consistency by evaluating Consistency Ratio (CR) for
13  validation of pairwise comparison matrix
14             iv. Defuzzification of the fuzzy weights
15  v. Normalize the crisp weights to measure criteria weights, sub-criteria weights
16  and global weights
17  }
```

---

**Algorithm 3:** CODAS (cr_weight[m], n)// CODAS method

---

```
1       {
2   // The following pseudocode will identify the Assessment Score (AS) based on
3   // which the ranking of the alternatives is obtained
4       Input: 'n' number of Alternatives, Criteria weight of each criteria and
5       sub-criteria
6       Output: Assessment Score (AS) of each Alternative
7       i. Construct the decision matrix of each alternatives based on criteria
8       ii. Obtain the normalized decision matrix
9       iii. Calculate the weighted normalized performance value
10      iv. Obtain the NIS point for each alternative
11  v. Calculate the Euclidean distance (ED) and Taxicab distance (TD) of
12  each alternative
13      vi. Create relative assessment matrix from ED and TD
14      vii. Calculate Assessment Score (AS) of each alternative
15      return AS[i];
16  }
```

---

**Algorithm 4:** MOORA (cr_weight[m], n)// MOORA method

---

```
1   {
2   // The following pseudocode will identify the Normalized Assessment Value
3      // (NAV) based on which the ranking of the alternatives is obtained
4       Input: 'n' number of Alternatives, Criteria weight of each criteria and
5       sub-criteria
6       Output:Normalized Assessment Value (NAV) of each Alternative
7   Create decision making matrix of all alternatives based on criteria available
8   Normalizing the decision matrix
9   Optimizing the attributes to find the normalized assessment value (NAV)
10  return NAV[i];
11      }
12
13
```

## 4. Methods

*4.1. Determination of Relevant Criteria and Their Associated Sub-Criteria in Choosing the mHealth Applications*

The criteria and their sub-criteria related to mHealth applications were carefully chosen. To choose the criteria and their sub-criteria in this research, a two-step procedure was used. Various criteria were identified in the early stages by reviewing the relevant literature. Following that review, expert opinions on the identified criteria were sought. Finally, a total of ten (10) criteria and twenty nine (29) sub-criteria [38] were selected as mentioned in Table 1 to evaluate all alternatives [38].

**Table 1.** Usability criteria and associated sub-criteria for assessing T2DM mHealth application.

| Criteria | Sub-Criteria |
|---|---|
| Learnability (A1) | Familiarity(A11) Learning time(A12) Minimal Action(A13) |
| Efficiency (A2) | No. of Taps(A21) Task Completion Rate(A22) Response Time(A23) Ease of Use(A24) Connection(A25) |
| Memorability (A3) | Saving(A31) Retain(A32) Reminder(A33) |
| Aesthetic (A4) | Attractive(A41) Appeal(A42) Organized(A43) |
| Error (A5) | Presence of Error(A51) |
| Navigation (A6) | Search(A61) Intuitive(A62) Involvement(A63) |
| Readability (A7) | Legible(A71) Understandable(A72) |
| Cognitive Load (A8) | Essentiality(A81) Presentation(A82) |
| Provision for Physically Challenged users (A9) | Weak Muscle Control(A91) Low Vision(A92) Hearing Impairment(A93) |
| Satisfaction (A10) | Provision(A101) Finding Correct Information(A102) Improvement(A103) Recommendation(A104) |

Once the criteria and sub-criteria were formulated, many mHealth applications, as mentioned in this research work, were investigated. The identified T2DM mHealth applications were used as the alternatives in this work.

The major goal of this methodology was to find the best T2DM mHealth applications. The criteria and their sub-criteria weights were computed by applying the FAHP in this model. In addition, CODAS and MOORA were used to rate the alternatives.

*4.2. Fuzzy AHP*

Fuzzy AHP is a powerful technique that considers uncertainty in human judgment. Two functions, the triangular membership function (TMF) and the trapezoidal membership function (TRMF), are most commonly used in FAHP. According to the literature,

various studies have employed TMF or TRMF to assess ambiguity and vagueness in expert judgments.

TMF is used in this study because of its popularity and its ease of computation. TMF turns expert-provided qualitative data into TFN. To overcome the uncertainties involved within the AHP methodology, the FAHP methodology, which is an extension of the traditional AHP method, utilized fuzzified comparison ratios that were described by TMF. The concept was to describe the weights of the nine-level judgment scales using triangular fuzzy numbers (TFNs) for depicting the relative significance of the criteria associated with the hierarchy [39]. Figure 2 depicts a TFN represented by the real numbers including *l*, *m*, and *u* and involving parameters such that *l< m< u*, where *l* is the least value, *m* indicates the promising value, and *u* is the maximum value related to the membership function $\mu_{\widetilde{A}}(x)$. The linguistic variables for indicating the relevance associated with each criterion are shown in Figure 3. The following equation represents a TFN's membership function:

$$\mu_{\widetilde{A}}(x) = \begin{cases} \frac{x-l}{m-l}, & l \leq x \leq m \\ \frac{u-x}{u-m}, & l \leq x \leq u \\ 0, & otherwise \end{cases} \tag{1}$$

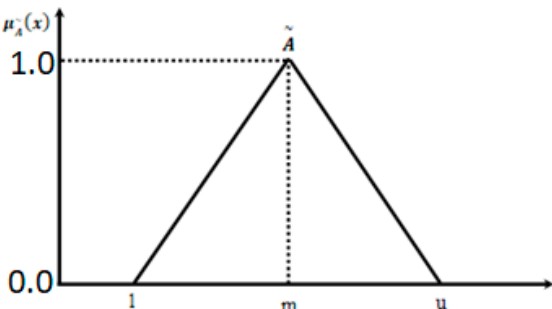

**Figure 2.** A triangular fuzzy number (TFN), $\widetilde{A}$.

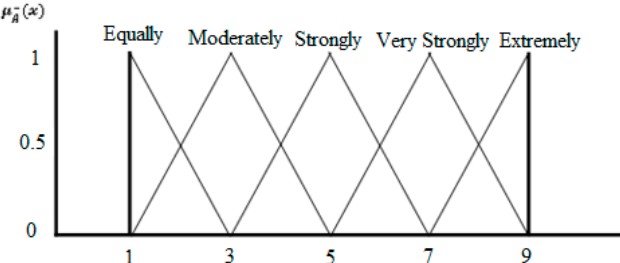

**Figure 3.** Linguistic variables used for determining importance weight of criteria.

The respective lower and upper limit of the fuzzy number $\widetilde{A}$ are represented by *l* and *u*, with *m* denoting the fuzzy number's mid-value.

Based on decision-maker (DM) assessments, the AHP approach was utilized to calculate criteria weights [40]. In the AHP method, which examines qualitative and quantitative criteria, pairwise comparisons are used. To deal with uncertainty issues, the regular AHP approach was expanded to include theory related to fuzzy sets, resulting in the FAHP. The essential steps employed in the FAHP are set out below:

**Step 1.** Generating the fuzzy pairwise comparison matrices

Fuzzy pairwise comparison matrices were created for each level, as illustrated in Table 2, utilizing crisp data acquired from experts on a well-defined fuzzy linguistic scale. Following that, using synthetic extent values, the extent analysis approach was employed to produce the priority weights. The fuzzy criterion assessment matrix was created by a pairwise comparison of the different attributes pertaining to the overall objective. The association between the TFN numerical values and linguistic characteristics were illustrated

by adopting the fuzzified Saaty's scale. When $\widetilde{A} = (l, m, u)$ was considered to be a TFN, the reciprocal of TFN, $A^{-1}$, was indicated as $(1/u_i, 1/m_i, 1/l_i)$.

**Table 2.** Fuzzy conversion scale [41].

| Linguistic Terms | Crisp Scale | TFS Scale | Reciprocal TFN Scale |
|---|---|---|---|
| Equally Preferred | 1 | (1,1,1) | (1/1,1/1,1/1) |
| Equally to moderately preferred | 2 | (1,2,3) | (1/3,1/2/,1/1) |
| Moderately preferred | 3 | (2,3,4) | (1/4,1/3,1/2) |
| Moderately to strongly preferred | 4 | (3,4,5) | (1/5,1/4,1/3) |
| Strongly preferred | 5 | (4,5,6) | (1/6,1/5,1/4) |
| Strongly to very strongly preferred | 6 | (5,6,7) | (1/7,1/6,1/5) |
| Very strongly preferred | 7 | (6,7,8) | (1/8,1/7,1/6) |
| Very strongly to extremely preferred | 8 | (7,8,9) | (1/9,1/8,1/7) |
| Extremely preferred | 9 | (8,9,9) | (1/9,1/9,1/8) |

The next goal was to develop a fuzzy pairwise comparison matrix. Decision-makers (DMs) use linguistic terms for developing a pairwise comparison matrix of criteria. To achieve this, we used the nine-point conversion scale developed by Anagnostopoulos et al. [41] to transform responses into fuzzy numbers (See Table 2).

The following is the resultant comparison matrix:

$$\widetilde{A} = \begin{bmatrix} 1 & \widetilde{a}_{12} & \dots & \widetilde{a}_{1n} \\ \widetilde{a}_{21} & 1 & \dots & \widetilde{a}_{2n} \\ \ddots & \ddots & \ddots & \ddots \\ \widetilde{a}_{m1} & \widetilde{a}_{m2} & \dots & 1 \end{bmatrix}$$

The aggregation of the fuzzy pairwise comparison matrix was completed. When it comes to group decision-making, the DMs' opinions are summarized as follows:

$$l_{ij} = \left(\prod_{k=1}^{k} l_{ijk}\right)^{1/k}, m_{ij} = \left(\prod_{k=1}^{k} m_{ijk}\right)^{1/k}, u_{ij} = \left(\prod_{k=1}^{k} u_{ijk}\right)^{1/k} \tag{2}$$

where, $\widetilde{A} = (l_{ij}, m_{ij}, u_{ij})$ and k indicates the number of DMs.

**Step 2.** Computation of normalized weight values using thegeometric mean (GM) method

We used the GM approach to compute normalized weights related to distinct criteria and their relevant sub-criteria. Due to the simplicity and ease in determining the highest Eigen value, and to reduce the judgment inconsistency, the GM technique was chosen.

The following were the steps involved in using the GM technique [42]:

*Calculation of Geometric Mean*: The geometric mean (GM) of the *j*th row associated with the fuzzy comparison matrix was calculated by the equation below.

$$\widetilde{GM}_i = \left[\prod_{i=1}^{n} \widetilde{a}_{ij}\right]^{1/n} \tag{3}$$

*Determination of normalized weight values*: The following equation gives the normalized weights for the *j*th row of the crisp comparison matrix:

$$\widetilde{W}_i = \frac{\widetilde{GM}_i}{\sum\limits_{i=1}^{n} \widetilde{GM}_i} \tag{4}$$

The fuzzy comparison values' geometric mean is represented by $\widetilde{GM}_i$, where the criteria weights are represented by $\widetilde{W}_i$.

**Step 3.** Check for consistency

Determination of the consistency ratio (*CR*) with respect to each pairwise comparison matrix was done to control the outcomes of the AHP technique, and the value had to be less than 0.1 to consider a matrix consistent.

Further, when a crisp comparison matrix was found to be consistent, the fuzzy pairwise comparison matrix was also in consistent state [43]. The equations below describe the consistency index (*CI*) and *CR* related to a comparison matrix:

$$CI = \frac{\lambda_{max} - n}{n - 1} \tag{5}$$

$$CR = \frac{CI}{RI} \tag{6}$$

where, $\lambda_{max} \rightarrow$ largest Eigen value, $n \rightarrow$ matrix size, and the standard value of *RI* is obtained using the matrix order.

**Step 4.** Defuzzification of the fuzzy weights

Because the weight is a fuzzy number, the center of area (COA) approach was used to defuzzify it, using the equation below:

$$W_i = \frac{(lw_i + mw_i + uw_i)}{3} \tag{7}$$

**Step 5.** Normalize the crisp weights

Crisp criteria weights were determined by normalizing the obtained crisp values via the following equation:

$$W_c = \frac{w_i}{\sum_{i=1}^{n} w_i} \tag{8}$$

### 4.3. Alternatives Used for the Study

We considered five mHealth applications (considered as alternatives) in evaluating usability and ranking them based on users' feedback as mentioned in Table 3 [38].

**Table 3.** Alternatives for the system under consideration.

| Alternatives | T2DM mHealth Applications |
|:---:|:---:|
| Alt 1 | Glucose Buddy |
| Alt 2 | mySugr |
| Alt 3 | Diabetes:M |
| Alt 4 | Blood Glucose Tracker |
| Alt 5 | OneTouch Reveal |

Figure 4 depicts the screenshots of the homepages of these applications. The services that these applications provide are listed below.

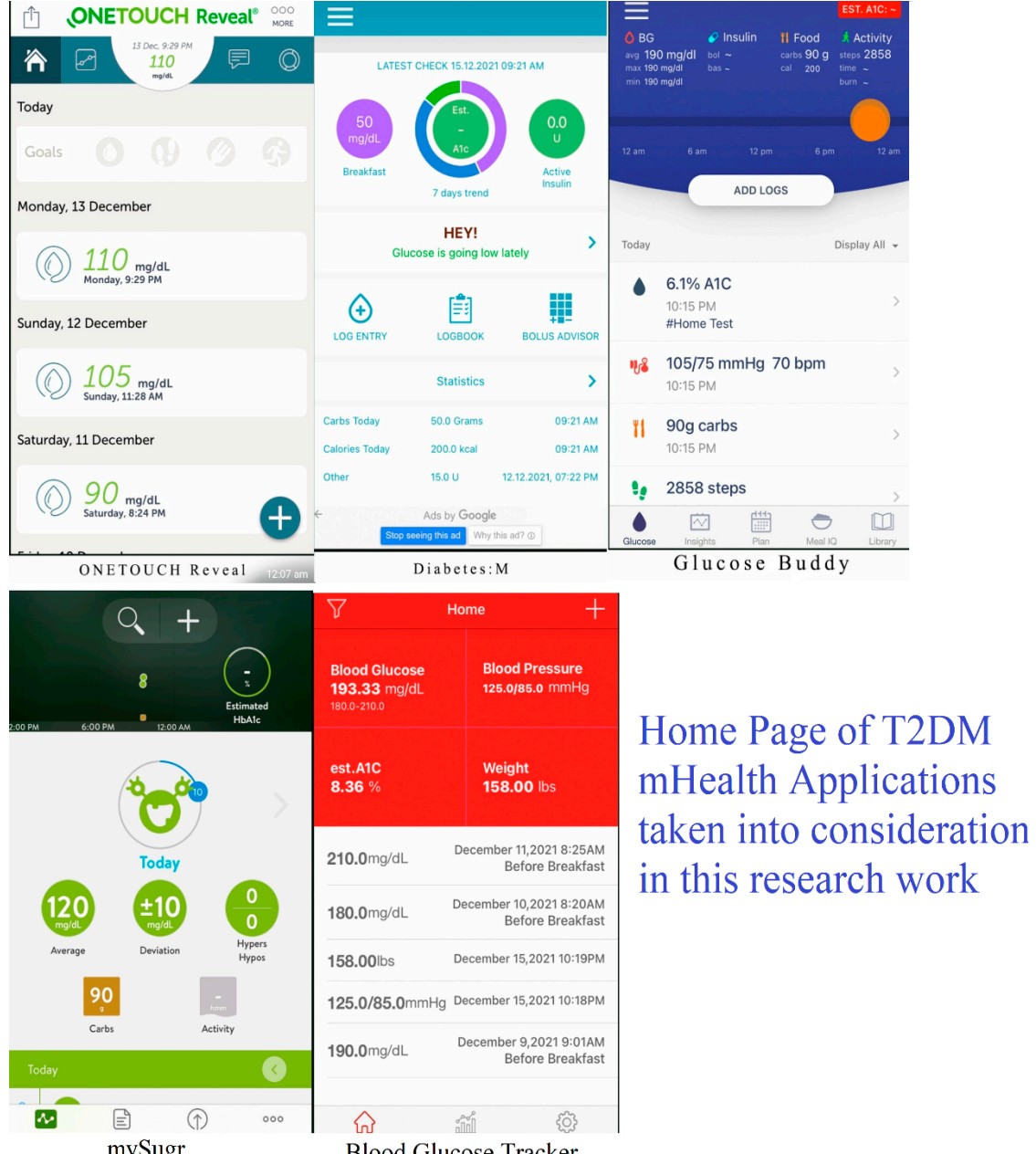

**Figure 4.** Screenshots of the homepage of the five T2DM mHealth applications taken into consideration.

Glucose Buddy (Alt1): This application provides the following services:

- It can simply enter blood glucose, medication, and meals in one entry, as well as track blood sugar, insulin, weight, blood pressure, A1C, and other trends.
- Notes to entries can be added for future reference, and can track walks and other cardio activities automatically.
- Real-time blood sugar monitoring is a simple and hassle-free solution for diabetes management.
- It offers professional assistance and guidance.

mySugr (Alt 2): The following are the best features of the mySugr application:

- It provides a personalized logging screen that can capture data from a Bluetooth-enabled blood glucose meter and it can analyze the trend to provide a quick overview of blood glucose levels.
- It provides a better searching feature for logging meals and activities, which aids with diabetes management.
- It has the capability to deliver the maximum level of data protection, in accordance with the general data protection regulation (GDPR).

Diabetes: M (Alt 3): By offering the following services, this mHealth application delivers everything necessary for optimal health management:

- It helps in providing the user with extensive information.
- It provides remote diabetes management that is effective.
- It displays the report in a statistical format (such as a bar chart) that helps users to understand it.
- It has the ability to recognize patterns and look for any pre-defined reoccurring problems, as well as the causes of their development.
- It incorporates an insulin bolus calculator that calculates insulin, depending on nutritional data.

Blood Glucose Tracker (Alt 4): The following are the services provided by this application:

- Throughout the day (such as breakfast, lunch, and supper), it monitors blood glucose at various levels to assist patients in maintaining effective blood sugar control.
- It can also track blood pressure, weight, and HbA1c levels, among other things.
- It filters history by event type/tag, where tags are useful to keep track of reactions to exercise, types of food, etc.

OneTouch Reveal (Alt 5): The following are the distinguishing characteristics of this mHealth application:

- It uses a unique color-coding system to organize blood sugar information in a way that novice users can understand.
- It provides automatically alerts for repeated highs or lows so that a user may take appropriate action.
- It establishes a daily goal for logging steps, carbohydrates, and activities.
- It notifies when it is time to take a blood sugar test and when it is time to take insulin.

### 4.4. Combinative Distance-Based Assessment (CODAS) Method

The alternatives' desirability is assessed by adopting two measures in this approach. The Euclidean distance (ED) between the alternatives and the negative ideal solution (NIS) are the most basic and important metrics. For criteria, the ED implies the usage of al2-norm indifference space. As a supplemental metric, the taxicab distance (TD) is used. The l1-norm indifference space (NIS) is related to it. The alternative that is farthest from the NIS is certainly the best. If the Euclidean distance cannot be used to compare two options, the taxicab distance can be used as a supplemental measure in this technique. Even though CODAS prefers the l2-norm indifference space, the method could also include two additional types of indifference spaces. If, for example, there are $n$ options and $m$ criteria from which we have to choose, the following steps are associated with this approach:

**Step 1.** Construct the decision-making matrix ($A$), as illustrated below:

$$A = [a_{ij}]_{m \times n} = \begin{bmatrix} a_{11} & a_{12} & \ldots & a_{1n} \\ a_{21} & a_{22} & \ldots & a_{2n} \\ \ddots & \ddots & \ddots & \ddots \\ a_{m1} & a_{m2} & \ldots & a_{mn} \end{bmatrix} \quad (9)$$

where $a_{ij}(a_{ij} \geq 0)$ signifies the $i$th alternative's performance value on the $j$th criterion and ($i \in \{1, 2, \ldots, m\}$ and $j \in \{1, 2, \ldots, n\}$).

**Step 2.** Obtain the normalized decision matrix. The following equation demonstrates the linear normalization in relation to the performance values:

$$N_{ij} = \begin{cases} \frac{a_{ij}}{\max\limits_{i} a_{ij}} & if \quad j \in C_b \\ \frac{\min\limits_{i} a_{ij}}{a_{ij}} & if \quad j \in C_{nb} \end{cases} \tag{10}$$

where, $C_b$ represents the benefit set and $C_{nb}$ refers to the cost criteria set.

**Step 3.** Calculate the normalised weighted decision matrix. The following formula is utilized to determine the weighted normalized performance values:

$$WN_{ij} = w_j N_{ij} \tag{11}$$

where, $w_j$ ($0 < w_j < 1$) indicates the weight related to the $j$th criterion, and $\sum\limits_{j=1}^{n} w_j = 1$

**Step 4.** Obtain the NIS point as follows:

$$NS = [NS_j]_{1 \times n} \tag{12}$$

$$NS_j = \min_{i} WN_{ij} \tag{13}$$

**Step 5.** Compute the ED and TD of the alternatives that are derived using negativeideal solutions, as shown below:

$$ED_i = \sqrt{\sum_{j=1}^{n} (WN_{ij} - NS_j)^2} \tag{14}$$

$$TD_i = \sum_{j=1}^{n} |WN_{ij} - NS_j| \tag{15}$$

**Step 6.** Create the relative assessment matrix, as shown below:

$$AM = [P_{ik}]_{n \times n} \tag{16}$$

$$P_{ik} = (ED_i - ED_k) + (\psi(ED_i - ED_k) \times (TD_i - TD_k)) \tag{17}$$

where, $k \in \{1, 2, \ldots, m\}$ and $\psi$ defines a threshold function for recognizing the equality of two alternatives' Euclidean distances. This threshold function is stated as follows:

$$\psi(a) = \begin{cases} 1 & if \quad |a| \geq \tau \\ 0 & if \quad |a| < \tau \end{cases} \tag{18}$$

The value $\tau$ denotes the threshold parameter specified by the decision-maker in this function. This option should be set between 0.01 and 0.05. When the difference in ED between two alternatives is smaller than $\tau$, the taxicab distance is employed to compare them. For performing the calculations, $\tau = 0.02$ has been taken in this study.

**Step 7.** Calculate each alternative's assessment score, using the equation below:

$$AS_i = \sum_{k=1}^{m} P_{ik} \tag{19}$$

**Step 8.** Rank the alternatives, depending on the order of decreasing assessment score $AS_i$ values. Among the alternatives, the option with the greatest $AS_i$ is the best choice.

### 4.5. Multi-Objective Optimization on the Basis of Ratio Analysis (MOORA) Method

MOORA is a strategy employed for the optimization of two or more conflicting qualities (objectives) at the same time, based on certain constraints. Multi-objective optimization challenges include maximizing profit while limiting product cost, optimizing vehicle performance while minimizing fuel consumption, and reducing weight thereby increasing a certain technical component strength [44]. The MOORA technique, proposed by Brauers [44], offers a multi-objective optimization methodology that is used to handle a variety of complicated decision-making challenges related to manufacturing. While ranking or selecting alternatives from a set of available options, the MOORA technique analyses both beneficial and non-beneficial criteria. The MOORA technique starts with the development of a decision matrix that compares the efficiency of several options in terms of numerous attributes (objectives). As a result, the MOORA approach can be efficiently used as a tool for rating and selecting alternatives from a large number of available options.

**Step 1.** Create a decision-making matrix including all of the available information related to the attributes. A Matrix $A_{m \times n}$ is used to represent the given data, as illustrated in the equation below.

$$A = [a_{ij}]_{m \times n} = \begin{bmatrix} a_{11} & a_{12} & \ldots & a_{1n} \\ a_{21} & a_{22} & \ldots & a_{2n} \\ \ddots & \ddots & \ddots & \ddots \\ a_{m1} & a_{m2} & \ldots & a_{mn} \end{bmatrix} \tag{20}$$

Here, $a_{ij}$ indicates the performance or effectiveness of the $i$th option associated with the $j$th attribute, m represents the count of alternatives present, and $n$ represents the count of attributes. The performance of each option on an attribute is then matched with a numerator that reflects all of the alternatives in regard to that attribute, resulting in a ratio system.

**Step 2.** Normalize the decision matrix

According to Brauers et al. [45], the square root of the sum of the squares of each alternative per attribute would be the best option for this denominator. It is expressed as follows:

$$a_{ij}^* = \frac{a_{ij}}{\sqrt{\left[\sum\limits_{i=1}^{m} x_{ij}^2\right]}} j = 1, 2, ..., n \tag{21}$$

where $a_{ij}^*$ refers to the normalized performance of the $i$th alternative based on the $j$th attribute. Here, $a_{ij}^*$ is a dimensionless number lying in the range [0, 1].

**Step 3.** Optimize the attributes and find the normalized assessment value

These normalized performances are summed up during maximization for the beneficial attributes and subtracted during minimization in multi-objective optimization for the non-beneficial attributes. As a result, the normalized assessment value ($NA_i$) is determined as follows:

$$NA_i = \sum\limits_{j=1}^{h} a_{ij}^* - \sum\limits_{j=h+1}^{n} a_{ij}^* \tag{22}$$

Here, $h$ represents the attributes to be maximized, (*n-h*) denotes the attributes that are to be minimized, and $NA_i$ indicates the normalized assessment value that is related to the $i$th alternative across the attributes. In numerous situations, it is quite common to find that some attributes seem to be more important than others. The weight of an attribute (significance coefficient) is multiplied by its corresponding weight (significance coefficient)

to give it additional prominence [46]. When the weights of these attributes are considered, the equation looks like this:

$$NA_i = \sum_{j=1}^{h} w_j a_{ij}^* - \sum_{j=h+1}^{n} w_j a_{ij}^* \tag{23}$$

where, $w_j$ is the $j$th attribute's weight, which is calculated by applying the AHP or the entropy approach.

**Step 4.** Rank the alternatives, depending on normalized assessment values

The $NA_i$ value might be positive or negative, based on the sum of the decision matrix's maxima that includes the beneficial attributes and the minimum value including the non-beneficial attributes. The final preference is shown by an ordinal ranking of $NA_i$. The best option will have the highest $NA_i$ value, while the worst option will have the lowest.

## 5. Result Analysis

### 5.1. Implementation of Pairwise Criteria Using the FAHP

Ten (10) major criteria and their twenty-nine (29) sub-criteria were determined for evaluating the usability aspect of mHealth applications by considering experts' judgments and the literature review. Crisp information about each criterion and sub-criteria was gathered, based on feedback from experts. By adopting a well-defined fuzzy linguistic Wang scale, the gathered crisp information for all the criteria was translated into TFN by following the scale mentioned in Table 2.

The fuzzy-based comparison decision matrix in relation to the criteria developed with TFN is shown in Table 4. The fuzzy comparison matrices in relation to all the associated sub-criteria are depicted in Tables 5–14 [38]. *CI* and *CR* were calculated using Step 4 of the FAHP to measure the consistency ratio. The *CR* values obtained for all criteria matrices are given in Table 15 [38]. Since, the *CR* value is found to be less than 0.1 for each criterion; it suggests that the pairwise comparison matrix created using expert feedback is consistent and suitable for further investigation.

**Table 4.** Fuzzy pairwise decision matrices for criteria.

| | Learnability | Efficiency | Memora-bility | Aesthetic | Error | Navigation | Readability | Cognitive Load | Provision for Physically Challenged Users | Satisfaction |
|---|---|---|---|---|---|---|---|---|---|---|
| Learnabil-ity | (1,1,1) | (1/4,1/3,1/2) | (1/5,1/4,1/3) | (1/4,1/3,1/2) | (1/4,1/3,1/2) | (1/4,1/3,1/2) | (1/4,1/3,1/2) | (1/4,1/3,1/2) | (1/5,1/4,1/3) | (1/4,1/3,1/2) |
| Efficiency | (2,3,4) | (1,1,1) | (3,4,5) | (1/3,1/2,1/1) | (1/3,1/2,1/1) | (1/3,1/2,1/1) | (1/4,1/3,1/2) | (1/3,1/2,1/1) | (1/4,1/3,1/2) | (1/4,1/3,1/2) |
| Memora-bility | (1/4,1/3,1/2) | (1/5,1/4,1/3) | (1,1,1) | (1/4,1/3,1/2) | (1/5,1/4,1/3) | (1/4,1/3,1/2) | (1/5,1/4,1/3) | (1/5,1/4,1/3) | (1/5,1/4,1/3) | (1/5,1/4,1/3) |
| Aesthetic | (3,4,5) | (1,2,3) | (2,3,4) | (1,1,1) | (1,2,3) | (2,3,4) | (2,3,4) | (2,3,4) | (1,2,3) | (1/3,1/2,1/1) |
| Error | (2,3,4) | (1,2,3) | (3,4,5) | (1/3,1/2,1/1) | (1,1,1) | (2,3,4) | (1,2,3) | (2,3,4) | (2,3,4) | (3,4,5) |
| Navigation | (2,3,4) | (1,2,3) | (2,3,4) | (1/4,1/3,1/2) | (1/4,1/3,1/2) | (1,1,1) | (1,2,3) | (1,2,3) | (1,2,3) | (2,3,4) |
| Readability | (2,3,4) | (2,3,4) | (3,4,5) | (1/4,1/3,1/2) | (1/3,1/2,1/1) | (1/3,1/2,1/1) | (1,1,1) | (2,3,4) | (1,2,3) | (1,2,3) |
| Cognitive Load | (2,3,4) | (1,2,3) | (3,4,5) | (1/4,1/3,1/2) | (1/4,1/3,1/2) | (1/3,1/2,1/1) | (1/4,1/3,1/2) | (1,1,1) | (3,4,5) | (1/3,1/2,1/1) |
| Provision for Physically Chal-lenged Users | (3,4,5) | (2,3,4) | (3,4,5) | (1/3,1/2,1/1) | (1/4,1/3,1/2) | (1/3,1/2,1/1) | (1/3,1/2,1/1) | (1/5,1/4,1/3) | (1,1,1) | (1/4,1/3,1/2) |
| Satisfaction | (2,3,4) | (2,3,4) | (3,4,5) | (1,2,3) | (1/5,1/4,1/3) | (1/4,1/3,1/2) | (1/3,1/2,1/1) | (1,2,3) | (2,3,4) | (1,1,1) |

**Table 5.** Fuzzy-based comparison matrix associated with the sub-criteria in relation to the learnability (A1) criteria.

| Learnability(A1) | Familiarity(A11) | Learning Time(A12) | Minimal Action(A13) |
|---|---|---|---|
| Familiarity(A11) | (1,1,1) | (2,3,4) | (1,2,3) |
| Learning time(A12) | (1/4,1/3,1/2) | (1,1,1) | (1/3,1/2,1/1) |
| Minimal Action(A13) | (1/3,1/2,1/1) | (1,2,3) | (1,1,1) |

**Table 6.** Fuzzy-based comparison matrix associated with the sub-criteria in relation to the efficiency (A2) criteria.

| Efficiency(A2) | No. of Taps(A21) | Task Completion Rate(A22) | Response Time(A23) | Ease of Use(A24) | Connection (A25) |
|---|---|---|---|---|---|
| **No. of Taps(A21)** | (1,1,1) | (1/3,1/2,1/1) | (1/4,1/3,1/2) | (1/6,1/5,1/4) | (1,2,3) |
| **Task Completion Rate(A22)** | (1,2,3) | (1,1,1) | (1,2,3) | (1/4,1/3,1/2) | (2,3,4) |
| **Response Time(A23)** | (2,3,4) | (1/3,1/2,1/1) | (1,1,1) | (1/3,1/2,1/1) | (1,2,3) |
| **Ease of Use(A24)** | (4,5,6) | (2,3,4) | (1,2,3) | (1,1,1) | (4,5,6) |
| **Connection(A25)** | (1/3,1/2,1/1) | (1/4,1/3,1/2) | (1/3,1/2,1/1) | (1/6,1/5,1/4) | (1,1,1) |

**Table 7.** Fuzz-based comparison matrix associated with the sub-criteria in relation to the memorability (A3) criteria.

| Memorability(A3) | Saving(A31) | Retain(A32) | Reminder(A33) |
|---|---|---|---|
| **Saving(A31)** | (1,1,1) | (2,3,4) | (2,3,4) |
| **Retain(A32)** | (1/4,1/3,1/2) | (1,1,1) | (2,3,4) |
| **Reminder(A33)** | (1/4,1/3,1/2) | (1/4,1/3,1/2) | (1,1,1) |

**Table 8.** Fuzzy-based comparison matrix associated with the sub-criteria in relation to the aesthetic (A4) criteria.

| Aesthetic(A4) | Attractive(A41) | Appeal(A42) | Organized(A43) |
|---|---|---|---|
| **Attractive(A41)** | (1,1,1) | (2,3,4) | (1/3) |
| **Appeal(A42)** | (1/4,1/3,1/2) | (1,1,1) | (1/5,1/4,1/3) |
| **Organized(A43)** | (2,3,4) | (3,4,5) | (1/4,1/3,1/2) |

**Table 9.** Fuzzy-based comparison matrix associated with the sub-criteria in relation to the error (A5) criteria.

| Error(A5) | Presence of Error(A51) |
|---|---|
| **Presence of Error(A51)** | (1,1,1) |

**Table 10.** Fuzzy-based comparison matrix associated with the sub-criteria in relation to the navigation (A6) criteria.

| Navigation(A6) | Search(A61) | Intuitive(A62) | Involvement(A63) |
|---|---|---|---|
| **Search(A61)** | (1,1,1) | (1/3,1/2,1/1) | (2,3,4) |
| **Intuitive(A62)** | (1,2,3) | (1,1,1) | (2,3,4) |
| **Involvement(A63)** | (1/4,1/3,1/2) | (1/4,1/3,1/2) | (1,1,1) |

**Table 11.** Fuzzy-based comparison matrix associated the sub-criteria in relation to the -readability (A7) criteria.

| Readability(A7) | Legible(A71) | Understandable(A72) |
|---|---|---|
| **Legible(A71)** | (1,1,1) | (2,3,4) |
| **Understandable(A72)** | (1/4,1/3,1/2) | (1,1,1) |

**Table 12.** Fuzzy based comparison matrix associated with the sub-criteria in relation to the cognitive load (A8) criteria.

| Cognitive Load(A8) | Essentiality(A81) | Presentation(A82) |
|---|---|---|
| **Essentiality(A81)** | (1,1,1) | (1/4,1/3,1/2) |
| **Presentation(A82)** | (2,3,4) | (1,1,1) |

**Table 13.** Fuzzy based comparison matrix associated with the sub-criteria in relation to the Provision for Physically Challenged Users (A9) criteria.

| Provision for Physically Challenged Users(A9) | Weak Muscle Control(A91) | Low Vision(A92) | Hearing Impairment(A93) |
|---|---|---|---|
| **Weak Muscle Control(A91)** | (1,1,1) | (1/5,1/4,1/3) | (1/4,1/3,1/2) |
| **Low Vision(A92)** | (3,4,5) | (1,1,1) | (1,2,3) |
| **Hearing Impairment(A93)** | (2,3,4) | (1/3,1/2,1/1) | (1,1,1) |

**Table 14.** Fuzzy-based comparison matrix associated with the sub-criteria in relation to the satisfaction (A10) criteria.

| Satisfaction(A10) | Provision(A101) | Finding Correct Information(A102) | Improvement(A103) | Recommendation(A104) |
|---|---|---|---|---|
| **Provision(A101)** | (1,1,1) | (1/4,1/3,1/2) | (1/3,1/2,1/1) | (1/5,1/4,1/3) |
| **Finding Correct Information(A102)** | (2,3,4) | (1,1,1) | (2,3,4) | (1/4,1/3,1/2) |
| **Improvement(A103)** | (1,2,3) | (1/4,1/3,1/2) | (1,1,1) | (1/4,1/3,1/2) |
| **Recommendation(A104)** | (3,4,5) | (2,3,4) | (2,3,4) | (1,1,1) |

**Table 15.** Consistency ratio (CR) for the criteria matrices.

| Criteria | Consistency Ratio (CR) |
|---|---|
| Learnability (A1) | 0.00477 |
| Efficiency (A2) | 0.04313 |
| Memorability (A3) | 0.09832 |
| Aesthetic (A4) | 0.06391 |
| Error (A5) | 0.00000 |
| Navigation (A6) | 0.04640 |
| Readability (A7) | 0.00000 |
| Cognitive Load (A8) | 0.00000 |
| Provision for Physically Challenged users (A9) | 0.01580 |
| Satisfaction (A10) | 0.05955 |

Each criterion and its sub-criteria weights were computed by applying the GM approach [42] once the comparison matrices were formed.

The weight computations for the sub-criteria were also completed, and their consistency was validated. Furthermore, as shown in Table 16, global weights were determined by multiplying the criteria weights by the sub-criteria weights.

**Table 16.** Weights of criteria, sub-criteria and global weights.

| Criteria | Sub-Criteria | Criteria Weight | Sub-Criteria Weights | Global Weights |
|---|---|---|---|---|
| **Learnability (A1)** | Familiarity(A11) | 0.0315 | 0.5392 | 0.0170 |
| | Learning time(A12) | | 0.1632 | 0.0051 |
| | Minimal Action(A13) | | 0.2974 | 0.0094 |
| **Efficiency (A2)** | No. of Taps(A21) | 0.0634 | 0.0962 | 0.0061 |
| | Task Completion Rate(A22) | | 0.2168 | 0.0137 |
| | Response Time(A23) | | 0.1807 | 0.0115 |
| | Ease of Use(A24) | | 0.4345 | 0.0276 |
| | Connection(A25) | | 0.0716 | 0.0045 |
| **Memorability (A3)** | Saving(A31) | 0.0280 | 0.5736 | 0.0160 |
| | Retain(A32) | | 0.2864 | 0.0080 |
| | Reminder(A33) | | 0.1399 | 0.0039 |
| **Aesthetic (A4)** | Attractive(A41) | 0.1712 | 0.2721 | 0.0466 |
| | Appeal(A42) | | 0.1199 | 0.0205 |
| | Organized(A43) | | 0.6080 | 0.1041 |
| **Error (A5)** | Presence of Error(A51) | 0.1849 | 1 | 0.1849 |
| **Navigation (A6)** | Search(A61) | 0.1225 | 0.3338 | 0.0409 |
| | Intuitive(A62) | | 0.5247 | 0.0643 |
| | Involvement(A63) | | 0.1416 | 0.0173 |
| **Readability (A7)** | Legible(A71) | 0.1229 | 0.75 | 0.0922 |
| | Understandable(A72) | | 0.25 | 0.0307 |
| **Cognitive Load (A8)** | Essentiality(A81) | 0.0869 | 0.25 | 0.0217 |
| | Presentation(A82) | | 0.75 | 0.0652 |
| **Provision for Physically Challenged users (A9)** | Weak Muscle Control(A91) | 0.0748 | 0.1226 | 0.0092 |
| | Low Vision(A92) | | 0.5571 | 0.0417 |
| | Hearing Impairment(A93) | | 0.3202 | 0.0240 |
| **Satisfaction (A10)** | Provision(A101) | 0.1138 | 0.0921 | 0.0105 |
| | Finding Correct Information(A102) | | 0.2720 | 0.0310 |
| | Improvement(A103) | | 0.1447 | 0.0165 |
| | Recommendation(A104) | | 0.4911 | 0.0559 |

*5.2. Ranking of mHealth Applications Based on Usability Aspect Using CODAS*

The weighted normalized decision matrix, NIS, ED, and TD of the alternatives given in Table 3 were calculated using Steps 1 to 5 referred to in Section 4.4 and are depicted in Table 17. Steps 6 and 7 from Section 4.4 were used to determine the relative assessment matrix and the alternative assessment scores.

**Table 17.** Weighted normalized decision matrix, NIS, ED and TD.

| Alternatives | Sub-Criteria | | | | | | | | | | | | | | | | Distances | |
|---|---|---|---|---|---|---|---|---|---|---|---|---|---|---|---|---|---|---|
| | A1.1 | A1.2 | A1.3 | A2.1 | A2.2 | A2.3 | A2.4 | A2.5 | … .. | A9.1 | A9.2 | A9.3 | A10.1 | A10.2 | A10.3 | A10.4 | ED | TD |
| **Alt 1** | 0.017 | 0.004 | 0.009 | 0.006 | 0.013 | 0.01 | 0.024 | 0.004 | … .. | 0.009 | 0.041 | 0.023 | 0.01 | 0.029 | 0.016 | 0.053 | 0.033 | 0.07 |
| **Alt 2** | 0.017 | 0.005 | 0.009 | 0.006 | 0.013 | 0.011 | 0.028 | 0.004 | … .. | 0.008 | 0.036 | 0.024 | 0.009 | 0.031 | 0.016 | 0.055 | 0.067 | 0.131 |
| **Alt 3** | 0.016 | 0.005 | 0.008 | 0.006 | 0.012 | 0.01 | 0.024 | 0.004 | … .. | 0.008 | 0.038 | 0.022 | 0.01 | 0.026 | 0.015 | 0.055 | 0.031 | 0.054 |
| **Alt 4** | 0.016 | 0.005 | 0.009 | 0.006 | 0.012 | 0.01 | 0.022 | 0.003 | … .. | 0.008 | 0.039 | 0.021 | 0.009 | 0.025 | 0.014 | 0.047 | 0.022 | 0.035 |
| **Alt 5** | 0.016 | 0.005 | 0.008 | 0.005 | 0.012 | 0.009 | 0.024 | 0.003 | … .. | 0.008 | 0.041 | 0.021 | 0.009 | 0.028 | 0.016 | 0.055 | 0.037 | 0.084 |
| **NIS** | 0.016 | 0.004 | 0.008 | 0.005 | 0.011 | 0.009 | 0.022 | 0.003 | … .. | 0.015 | 0.042 | 0.008 | 0.036 | 0.021 | 0.009 | 0.025 | 0.014 | 0.047 |

Based on the assessment score obtained in Table 18, Alt 2 (mySugr) had the highest assessment score ($AS_i$) (mySugr),while Alt 4 (Blood Glucose Tracker) had the lowest assessment score, indicating that mySugr is the best mHealth application in respect of the usability parameter. The ranking of the mHealth applications obtained using the CODAS approach based on the $AS_i$ value is shown in Figure 5.

**Table 18.** Relative assessment matrix and assessment scores in relation to the alternatives.

| | Alt 1 | Alt 2 | Alt 3 | Alt 4 | Alt 5 | $AS_i$ (Assessment Scores) |
|---|---|---|---|---|---|---|
| **Alt 1** | 0.0000 | −0.0343 | 0.0017 | 0.0106 | −0.0050 | −0.0270 |
| **Alt 2** | 0.0344 | 0.0000 | 0.0361 | 0.0450 | 0.0295 | **0.1450** |
| **Alt 3** | −0.0017 | −0.0360 | 0.0000 | 0.0089 | −0.0066 | −0.0355 |
| **Alt 4** | −0.0105 | −0.0449 | −0.0089 | 0.0000 | −0.0155 | −0.0797 |
| **Alt 5** | 0.0050 | −0.0294 | 0.0067 | 0.0155 | 0.0000 | −0.0023 |

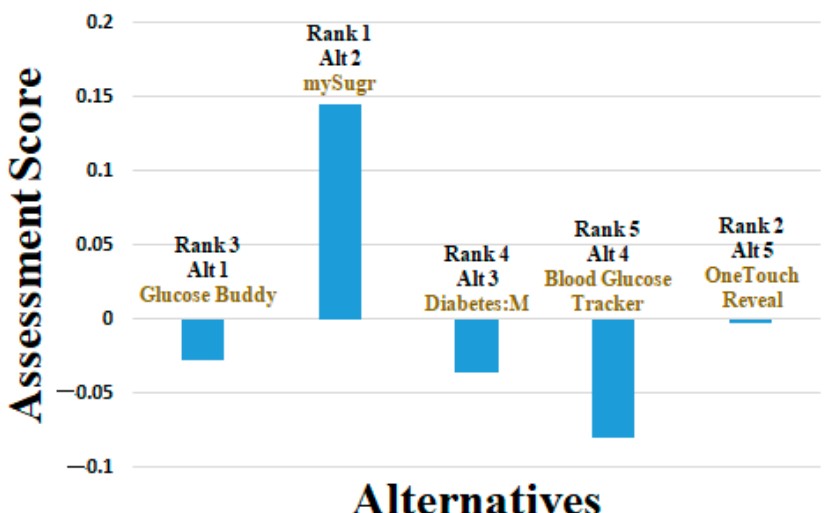

**Figure 5.** Rank obtained by CODAS based on assessment score.

To assess the proposed framework's stability and validity, a sensitivity analysis was performed. Threshold values ranged from 0.01 to 0.05, with assessment score ($AS_i$) values recorded for each threshold value. Similar rankings were produced for each threshold value using the indicated $AS_i$ values, showing that the ranking results are stable and valid. Figure 6 shows the sensitivity analysis-based ranking achieved at various τ values.

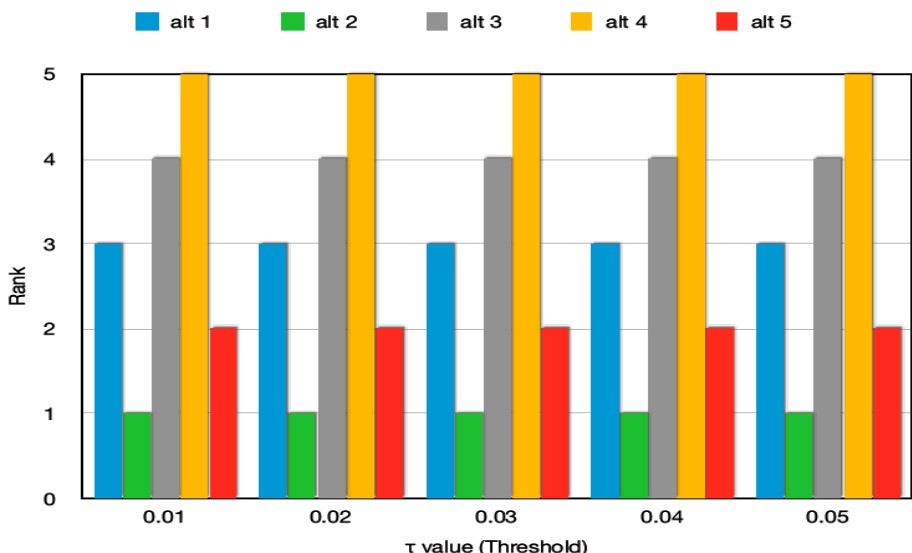

**Figure 6.** Ranking results produced for various τ values based on sensitivity analysis.

### 5.3. Ranking of mHealth Applications Based on Usability Aspect Using MOORA

The weighted normalized decision matrix and normalized assessment ($NA_i$) values were obtained by following Steps 1 to 4 of MOORA, as mentioned in Section 4.4 and displayed in Table 19. The normalized assessment ($NA_i$) was employed to determine the rank. The option with the highest $NA_i$ value was the best. Figure 7 depicts the ranking given to the alternatives on the basis of the values of $NA_i$. According to the MOORA approach, mySugr was the best application and the Blood Glucose Tracker was the lowest ranked of the five applications available.

**Table 19.** Weighted normalized decision matrix and normalized assessment ($NA_i$) values.

| Alternatives | Sub-Criteria | | | | | | | | | | | | | | | | Normalized Assessment Values |
|---|---|---|---|---|---|---|---|---|---|---|---|---|---|---|---|---|---|
| | A1.1 | A1.2 | A1.3 | A2.1 | A2.2 | A2.3 | A2.4 | A2.5 | … .. | A9.1 | A9.2 | A9.3 | A10.1 | A10.2 | A10.3 | A10.4 | |
| **Alt 1** | 0.0076 | 0.002 | 0.0043 | 0.0028 | 0.0063 | 0.0049 | 0.0119 | 0.002 | … .. | 0.0082 | 0.008 | 0.0078 | 0.008 | 0.008 | 0.0077 | 0.0076 | 0.055989 |
| **Alt 2** | 0.0079 | 0.0024 | 0.0045 | 0.0028 | 0.0068 | 0.0057 | 0.0139 | 0.0024 | … .. | 0.0074 | 0.0071 | 0.0081 | 0.0073 | 0.0083 | 0.0079 | 0.0079 | **0.093095** |
| **Alt 3** | 0.0076 | 0.0024 | 0.004 | 0.0026 | 0.0061 | 0.0052 | 0.0121 | 0.0019 | … .. | 0.0077 | 0.0074 | 0.0075 | 0.0078 | 0.0072 | 0.0075 | 0.0079 | 0.05649 |
| **Alt 4** | 0.0075 | 0.0023 | 0.0041 | 0.0028 | 0.0058 | 0.0052 | 0.0113 | 0.0018 | … .. | 0.0074 | 0.0076 | 0.0072 | 0.0075 | 0.0068 | 0.007 | 0.0067 | 0.048087 |
| **Alt 5** | 0.0075 | 0.0024 | 0.004 | 0.0025 | 0.0058 | 0.0045 | 0.0121 | 0.002 | … .. | 0.0074 | 0.0079 | 0.0073 | 0.0073 | 0.0077 | 0.0078 | 0.0078 | 0.067588 |

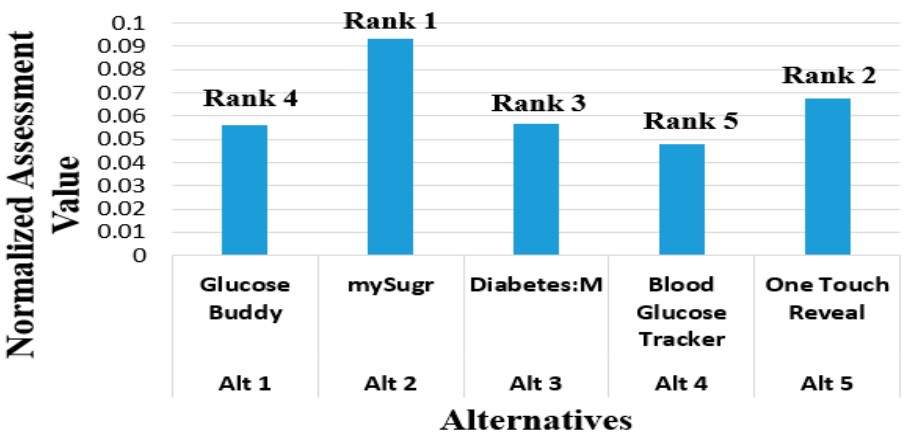

**Figure 7.** Rank obtained by MOORA based on normalized assessment value ($NA_i$).

*5.4. Comparison and Validation of the Results Obtained by CODAS-FAHP and MOORA-FAHP*

This study identified five (05) mHealth applications for evaluating usability criteria and these applications are useful for self-management of T2DM patients. Self-management of diabetes-related chronic disease is very essential in maintaining healthy lifestyles and reduces suffering.

In this paper, we proposed two models (MOORA and CODAS) to evaluate the usability of mHealth applications and to rank them based on ten criteria and twenty-nine sub-criteria. The FAHP was used to identify the consistent criteria and sub-criteria. The CODAS method measured the Euclidean distance (ED) and the negative ideal solution (NIS) for determining usability. It has been observed that the alternative farthest from the NIS is the best mHealth application. Ranking of the applications was carried out based on assessment scores. The MOORA method was the multi-objective optimization method which analyzed both beneficial and non-beneficial criteria for ranking the mHealth applications. That method measured normalized assessment ($NA_i$) value and showed that the application with the highest $NA_i$ value was the best mHealth application.

Comparing the results of CODAS and MOORA, among the T2DM mHealth applications included in this study, mySugr was the best-ranked application, while the Blood Glucose Tracker was the least desired application. The rankings determined by CODAS and MOORA were slightly different. In CODAS, Glucose Buddy had Rank 3 and Diabetes: M had Rank 4, whereas in MOORA, Glucose Buddy had Rank 4 and Diabetes: M had Rank 3. The compared results of the rankings of the alternatives using these two methodologies is depicted in Figure 8.

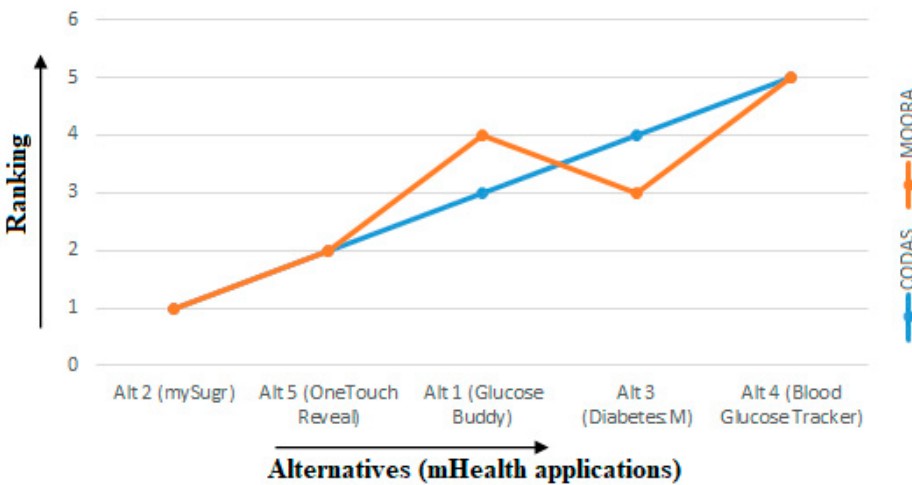

**Figure 8.** Comparison result of the rankings of the alternatives using CODAS and MOORA.

**6. Conclusions**

Because of increased digitization, mHealth has become necessary for regular healthcare and monitoring. With the growing number of health-related applications, determining which applications are appropriate for users is becoming increasingly complex. This study contributes by proposing novel hybrid MCDM models for choosing the best T2DM mHealth applications by evaluating usability and user interface. The authors presented hybrid decision-making models by utilizing the FAHP on CODAS and on MOORA to assess the usability aspect of the applications for efficient selection on the basis of multiple criteria and sub-criteria. The FAHP in combination with CODAS and MOORA produced appropriate results, and will aid decision-makers in seeing how different criteria affect final conclusions. The FAHP helped to reduce vagueness and errors obtained from expert judgments, while CODAS and MOORA assessed the usability parameters of the mHealth applications and provided ranking by considering several criteria and sub-criteria. Among the five alternatives, mySugr emerged as the best mHealth application, while the Blood Glucose

Tracker was the least preferred. The result seemed to be consistent in both approaches. The stability and validity of the findings were tested using sensitivity analysis. The sensitivity results verified the proposed frameworks' stability.

The findings of this study can aid patients and clinicians in making strategic and tactical decisions about which T2DM mHealth application to use. Furthermore, this research may be expanded to establish an environment for selecting appropriate T2DM mHealth applications and, in particular, to extend the models for T2DM mHealth application evaluation and selection.

**Author Contributions:** Conceptualization, K.G.; methodology, S.R.; software, R.C.P.; validation, A.A. and R.K.; formal analysis, S.R.N.; investigation; resources, A.K.J.S.; data duration, A.A.; writing—review and editing, A.K.J.S.; visualization, K.G.; supervision, R.C.P.; project administration, R.C.P.; funding acquisition, S.R., R.K. All authors have read and agreed to the published version of the manuscript.

**Funding:** Project number (RSP2022R498), King Saud University, Riyadh, Saudi Arabia.

**Institutional Review Board Statement:** Not applicable.

**Informed Consent Statement:** Not applicable.

**Data Availability Statement:** Data sharing not applicable to this article as no datasets were generated or analyzed during the current study.

**Acknowledgments:** Researchers supporting project number (RSP2022R498), King Saud University, Riyadh, Saudi Arabia.

**Conflicts of Interest:** The authors declare no conflict of interest.

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
