# Peer review of "Multi-Criteria Usability Evaluation of mHealth Applications on Type 2 Diabetes Mellitus Using Two Hybrid MCDM Models: CODAS-FAHP and MOORA-FAHP"

_applsci, doi:10.3390/app12094156_

Round 1

Reviewer 1 Report

The article proposes a hybrid MCDM model for choosing the best T2DM mHealth applications by using fuzzy AHP in combination with CODAS and MOORA methods.

Last paragraph of section 1: revise last sentence “Eventually, Section 6 discusses the conclusion”. Change “Eventually” to a more adequate word, like “Lastly” or similar.

For better structure of the paper, the sentences in section 3 could be included inside section 1.

Page 7, line 270, reference to Fig. 2 that does not appears in the paper. Line 311 also refers to Fig. 2.

The mathematical expressions included in the text should have the same formatting as the corresponding equations. We found this problem throughout the paper.

Line 360, repeated sentence.

Section 5 presents several procedures for each of the applied methods. Due to the numerous steps involved, sometimes its difficult to track all of them. Try to summarize some (or all) of the methods through a pseudo-code.

Section 6 shows that the rank of mHealth applications using the CODAS and MOORA results in the same best mHealth, that is, mySugr. For other tested applications the results differ. The rank of the best application could not be different (for this or other applications)? If so, the utility of the methods is not useless? Comment on this. How the proposed methods compare with other similar techniques?

Add more updated references.

Author Response

Last paragraph of section 1: revise last sentence “Eventually, Section 6 discusses the conclusion”. Change “Eventually” to a more adequate word, like “Lastly” or similar.

Response: The changes has been incorporated in Section 1 of the revised manuscript.

For better structure of the paper, the sentences in section 3 could be included inside section 1.

Response: The aim of the paper mentioned in Section 3 has been included inside section 1 as reflected in the revised manuscript.

Page 7, line 270, reference to Fig. 2 that does not appears in the paper. Line 311 also refers to Fig. 2.

Response: Comment is not clear to the authors, However Figure 2 shows the Triangular Fuzzy Number (TFN) in page no. 10 of the revised manuscript, under Section 4.2 (Fuzzy AHP)

The mathematical expressions included in the text should have the same formatting as the corresponding equations. We found this problem throughout the paper.

Response: Mathematical expressions are all modified and is reflected in the revised manuscript. However, the manuscript should be saved in .docx format.

Line 360, repeated sentence.

Response: Sorry, the comment is not clear to the authors, However, It has been modified and reflected in the revised manuscript.

Section 5 presents several procedures for each of the applied methods. Due to the numerous steps involved, sometimes its difficult to track all of them. Try to summarize some (or all) of the methods through a pseudo-code.

Response: The pseudo-code is presented under Section 3 (Proposed Hybrid Model) of the revised manuscript, which includes all the methods used in the proposed hybrid model.

Section 6 shows that the rank of mHealth applications using the CODAS and MOORA results in the same best mHealth, that is, mySugr. For other tested applications the results differ. The rank of the best application could not be different (for this or other applications)? If so, the utility of the methods is not useless? Comment on this. How the proposed methods compare with other similar techniques?

Response: From the result analysis it has been observed that for both the methods (CODAS and MOORA), mySugr is the best mHealth application in terms of usability aspect. There is a slight variation for only two mHealth applications i.e. Glucose Buddy and Diabetes:M (Incase of CODAS method, Glucose Buddy ranks 3rd and Diabetes:M ranks 4th, whereas incase of MOORA, Glucose Buddy ranks 4th and Diabetes:M ranks 3rd). This is because of the fact that these two applications are having very few changes in their features in terms of usability aspect, which make very slight modifications in their ranking.

Add more updated references.

Response: Updated references has been incorporated in Reference section and reflected in Section 2 (Related Works) of the revised manuscript. The references added in Related works are numbered from [32] to [36]

Reviewer 2 Report

  1. Figure 01 has low quality. Please increase the resolution of this image.
  2. Mathematical formulas have very poor quality. Please type them using Microsoft Word Equation or Math Type.
  3. Please provide more details about alternative mHealth applications (i.e., Glucose Buddy, mySugr, ...). What are their features? What are their pros and cons?
  4. Please bold the best result in each Table for more reader attention.
  5. In Figure 04, the chart and axis labels are overlapped.
  6. The overall structure of the paper is appropriate. However, it would be better if section 3 goes at the end of the Introduction section.
  7. Please provide more discussion on the results.
  8. Most of the references used in the paper are outdated. Please replace references with new ones, and mention newer articles in the related works section.
  9. References do not have the same format. For example [30] and [41].

Author Response

2. Figure 01 has low quality. Please increase the resolution of this image.

Response: Resolution of Figure 1 has been increased and reflected in the revised manuscript under Section 3.

2. Mathematical formulas have very poor quality. Please type them using Microsoft Word Equation or Math Type.

Response: Mathematical formulas are all modified using Word Equation and is reflected in the revised manuscript. However, the manuscript has to be save in .docx format.

3. Please provide more details about alternative mHealth applications (i.e., Glucose Buddy, mySugr, ...). What are their features? What are their pros and cons?

Response: The details of all five (05) alternatives i.e. mHealth applications along with their features are incorporated under Section 4.3 (Alternatives used for the study) of the revised manuscript.

4. Please bold the best result in each Table for more reader attention.

Response: Best result is highlighted in bold in Table 18 & 19 of the revised manuscript.

5. In Figure 04, the chart and axis labels are overlapped.

Response: The figure has been modified and numbered as Figure 5 in the revised manuscript.

6. The overall structure of the paper is appropriate. However, it would be better if section 3 goes at the end of the Introduction section.

Response: The aim of the paper mentioned in Section 3 has been included inside section 1 as reflected in the revised manuscript.

7. Please provide more discussion on the results.

Response: Discussion on the results obtained by the proposed method is reflected under section 5.4 (Comparison and validation of the result obtained by CODAS-FAHP and MOORA-FAHP) of the revised manuscript.

8. Most of the references used in the paper are outdated. Please replace references with new ones, and mention newer articles in the related works section.

Response: Latest/ Updated references has been incorporated in Reference section and reflected in Section 2 (Related Works) of the revised manuscript. The references added in Related works are numbered from [32] to [36]

9. References do not have the same format. For example [30] and [41].

Response: All the references are changed to same format and are reflected in the Reference section of the revised manuscript.